# Antigen heterogeneity in the development and clinical validation of a multiplexed urine test for tuberculosis

Tyler J. Dougan [1,2,3,4,5,6], Shira Roth [1,2,3,6], Liangxia Xie [1,2,3], Sydney D'Amaddio[1,2,3] &
David R. Walt [1,2,3] ✉

## Abstract

**Background** Tuberculosis (TB) is one of the leading causes of death worldwide, even though it is curable using antibiotics. Most people who die of TB never begin treatment because diagnostics are insufficiently sensitive and accessible. We aimed to measure low-abundance biomarkers and diagnose TB in urine.
**Methods** We developed and clinically validated a multiplex Single Molecule Array (Simoa) assay to detect TB in urine by measuring two TB biomarkers: lipoarabinomannan (LAM) and antigen 85B (Ag85B). Using antibodies that recognize different epitopes of LAM in a four-plex assay with three LAM and one Ag85B antibody pairs, we trained a model and demonstrated its performance in retrospective cohorts totaling 576 individuals from South Africa, Peru, Vietnam, and Cambodia, including a blinded test cohort ($n = 215$).
**Results** Here we present an assay that classifies samples with 98% specificity, 45% sensitivity overall, and 58% sensitivity among people living with the human immunodeficiency virus (HIV).
**Conclusions** Different antibody pairs detecting different epitopes on LAM report diverging concentrations. We do not find that adding antibody pairs to detect different epitopes on LAM improves the assay's accuracy. Our assay is more sensitive than the existing AlereLAM lateral flow test for TB in HIV-positive individuals, uses safe and accessible urine samples, and represents a step towards an adjunctive diagnostic test to aid clinicians in starting treatment.

## Plain Language Summary

The bacterial disease tuberculosis (TB) kills over one million people each year despite being curable, largely because of limitations in clinicians' ability to easily diagnose it. We developed a urine-based test to measure tuberculosis-associated molecules using a more sensitive assay to determine if we could improve upon the clinical sensitivity and specificity of existing tests. We tested 576 urine samples with assays that detect different regions of two different TB-derived molecules: lipoarabinomannan (LAM) and antigen 85B. Our test correctly identified 98% of people without TB and 45% of TB cases overall, including 58% among people with HIV, higher than the existing test that detects LAM. We found that LAM measurements change depending on which assays are used, but combining these assays did not meaningfully improve accuracy. This work moves us closer to a safe, non-invasive urine test to help doctors start TB treatment sooner.

Tuberculosis (TB) is the leading cause of death from a single infectious agent, killing over one million people each year[1]. It is caused by the bacillus *Mycobacterium tuberculosis* (*M. tb*)[1]. Although TB disease is curable with 6 months to 2 years of antibiotics, most people who die of TB never begin treatment[1] because diagnostics are insufficiently sensitive and accessible. About 60% of TB diagnoses are made with culture, smear microscopy, and nucleic acid amplification tests (NAATs)[1]. These tests provide bacteriological confirmation of TB, which is pathogen-specific and enables definitive diagnosis, case registration, and drug resistance testing[1]. However,

bacteriological culture cannot be a primary test because it takes weeks to return a result[2]. Smear microscopy fails to detect about half of TB cases[3]. Cepheid Xpert, MolBio Truenat and other molecular tests are faster and more sensitive, and rely on sputum, which has much higher loads of detectable bacilli than other specimen types. However, sputum from people with pulmonary TB is highly infectious[4]. In addition, some patients with TB symptoms (especially children, people living with HIV, and people with extrapulmonary TB) cannot produce enough sputum for testing[5]. The requirements for conducting these procedures safely are a major driver of

[1]Wyss Institute for Biologically Inspired Engineering, Harvard University, Boston, MA, USA. [2]Department of Pathology, Brigham and Women's Hospital, Boston, MA, USA. [3]Harvard Medical School, Harvard University, Boston, MA, USA. [4]Harvard-MIT Program in Health Sciences and Technology, Massachusetts Institute of Technology, Cambridge, MA, USA. [5]Present address: Africa Health Research Institute, Durban, South Africa. [6]These authors contributed equally: Tyler J. Dougan, Shira Roth. ✉e-mail: dwalt@bwh.harvard.edu

the cost of TB diagnostic clinics and laboratories, limiting their expansion and accessibility[6].

The World Health Organization (WHO) has approved one non-sputum-based test: the Abbott (formerly Alere) Determine™ TB LAM Ag test (AlereLAM)[5]. AlereLAM is a lateral flow assay for the *M. tb* antigen lipoarabinomannan (LAM) in urine that gives results in 25 minutes[7]. However, it is only recommended for use in people with HIV, who comprise 6% of TB cases; even then, it is only 42% sensitive and 91% specific[5,7].

In 2014, the WHO released target product profiles (TPPs) for four urgently needed TB diagnostics, including a rapid biomarker-based non-sputum-based test for detecting TB[8]. Ten years later, this unmet need was updated with a TPP on a rapid test for detecting *M. tuberculosis* at the peripheral level[9], with emphasis on test complexity (POC/near POC/low complexity) and sample type (sputum/non-sputum).

Urine offers particular advantages as a non-sputum sample: it is plentiful and simple to collect from adults and children without generating hazardous bioaerosols[10]. The glycolipid LAM is the TB biomarker with the strongest supporting evidence and is the target of the AlereLAM assay[11]. LAM is released from active or degrading bacilli, enters the bloodstream where it associates with lipoproteins, is filtered by the kidneys, and appears in urine. It is detectable in sputum and urine, but present at much lower levels in serum unless aggressive extraction is used[10]. Urinary LAM levels vary widely and are typically highest in people living with HIV and in advanced, disseminated, or renal TB[10].

Structurally, LAM ( ~ 17 kDa) consists of a phospholipid anchor linked to a conserved mannose core, from which an arabinan domain extends with variable side chains and mannose caps[10]. Variation in these caps generates diverse LAM isoforms[10]. Because epitopes are repeated multiple times[12], LAM can be detected using antibodies targeting different epitopes: S4-20 (Man2/Man3 caps with MTX)[11], FIND28 (Ara6 with or without Man caps)[11], A194-01 (Ara4/Ara6 with or without Man1)[12,13], and G3 (Man2/Man3)[11,14].

*M. tb* also has a number of promising protein biomarkers, including the diacylglycerol acyltransferase/mycolyltransferase antigen 85 complex of fibronectin-binding proteins (Ag85, *fbpB*)[15]. Ag85B is one of the most antigenic proteins of *M. tb*[16].

There is a strong association between analytical sensitivity (limit of detection, LOD) and clinical sensitivity in LAM tests. For example, AlereLAM's LOD is not reported, but it is likely ~1 ng/mL[10] and its clinical sensitivity is low (42% sensitivity)[7]. Paris et al.[17,18] showed that LAM was detectable in urine from HIV-negative people with TB using an assay with an LOD of 14 pg/mL. Sigal et al.[11] compared many antibody pairs in a sandwich immunoassay format (Meso Scale Discovery, sometimes referred to as MSD or EclLAM due to its electrochemiluminescent readout). They found that two pairs of monoclonal antibodies (S4-20/A194-01 and FIND28/A194-01) yielded assays with very low LODs (6 and 11 pg/mL, respectively) and high clinical sensitivity in smear-positive TB[11]. As the reported concentrations of LAM are in the pg/mL order of magnitude and the current AlereLAM and EcLAM are not sensitive enough, we aimed to develop an ultrasensitive assay for LAM.

Many tools can test for proteins and similar macromolecules, but single-molecule arrays (Simoa) are ideal for detecting TB in diverse patients because they can quantify analytes present at very low concentrations[19]. A Simoa assay has been developed to measure LAM in serum with high analytical sensitivity but low clinical sensitivity, likely due to shielding by lipoproteins as discussed above[20]. In another study, Simoa was used to measure four cytokines and immunoglobulins to one *M. tb* protein, Ag85B, in serum and plasma as a triage test[21].

Our hypothesis was that recognizing different epitopes on LAM would improve the test accuracy. We report the development and validation of a Simoa assay to diagnose TB by measuring both LAM and Ag85B in urine. We exploited antibodies that recognize different epitopes on LAM to develop a four-plex assay with three pairs of antibodies measuring LAM and one pair measuring Ag85B. Across 576 samples from four countries, our assay detected TB with 98% specificity, 45% sensitivity overall, and 58%

sensitivity among people living with the human immunodeficiency virus (HIV), modestly outperforming AlereLAM. This multiplexed approach yields equivocal gains in accuracy but demonstrates how LAM forms and concentrations vary, and clarifies the opportunities and limits of future LAM tests.

## Methods

### Antibodies and standards

The following reagents were obtained through BEI Resources, NIAID, NIH: *M. tb*, Strain H37Rv, Purified LAM, NR-14848 and Ag85B (gene Rv1886c), Purified Native Protein from *M. tb*, Strain H37Rv, NR-53526. FIND28 antibody was obtained from the Foundation for Innovative New Diagnostics (FIND; Geneva, Switzerland). G3 and S4-20 antibodies were kindly provided by Otsuka Pharmaceutical Co. (Tokyo, Japan). A194-01 IgM detection antibody for LAM was provided by Rutgers University (Newark, NJ). Ag85B capture (182λ) and detection (149κ) antibodies were obtained from AbCellera Biologics (Vancouver, Canada) under material transfer agreements. GenScript was contracted to produce large batches of antibodies using sequence information provided by AbCellera Biologics under non-disclosure agreements.

### Study design

Assays were developed for 11 *M. tb* antigens in urine and tested in a discovery cohort (Tables S1–S3, and Figs. S1–S2, SI). Only LAM and Ag85B were detectable and showed significant differences between TB and non-TB patients. Therefore, these markers were selected for further optimization. After development and validation, this multiplexed assay for LAM and Ag85B was evaluated in a retrospective case-control study to build and evaluate a diagnostic model for TB.

This study was determined not to constitute human subjects research by the Partners Human Research Committee (Protocol #2017P001447) as it involved only secondary analysis of de-identified samples. All urine samples were provided by the FIND TB sample repository. FIND collected these samples under institutional review board (IRB)/independent ethics committee (IEC) approved studies in participating countries. Urine samples were collected from adult subjects with symptoms suggestive of pulmonary TB in South Africa, Peru, Vietnam, and Cambodia between June 2012 and February 2019, and all samples were collected prior to medical intervention. Informed consent was obtained from all participants at enrollment, and no personally identifiable information was available to researchers. All clinical data were anonymized, and barcodes were used to access pertinent information. FIND uses standardized protocols for collecting and processing samples, which were reported previously in detail[11].

The study consisted of three cohorts: a model-building (training) cohort of 120 participants, a validation cohort of 241 participants, and a blinded test cohort of 215 participants (Table 1). Samples were selected retrospectively from FIND's specimen bank based on HIV status and TB category. For the latter, smear positivity (S ± ), culture positivity (C ± ), latent TB infection (LTBI), and signs and symptoms led to six categories: smear-positive TB (S + C + TB), smear-negative culture-positive TB (S–C + TB), clinically diagnosed TB, non-TB/non-LTBI, non-TB/LTBI, and likely subclinical TB. Clinically diagnosed and likely subclinical TB cases were smear- and culture-negative; clinically diagnosed patients were started on empirical treatment, while likely subclinical patients were not started on treatment but were culture-positive at a follow-up visit 44–175 days later. (Unlike most definitions of "subclinical TB" in the literature, FIND's "likely subclinical TB" cases are symptomatic but bacteriologically negative.) Following the decision to treat, we classified clinically diagnosed cases as TB and likely subclinical cases as non-TB (Table S5). "Likely subclinical TB" cases were classified as non-TB because all were bacteriologically negative at enrollment by smear ( ≥ 2), solid culture ( ≥ 2), liquid culture ( ≥ 2), and Xpert ( ≥ 1); we cannot distinguish baseline culture-negative TB from incident infection acquired after enrollment. Cohorts were procured sequentially: (1) a convenience cohort of 100 samples to establish performance; (2) a validation cohort of 258 procured based on a power calculation

## Table 1 | Clinical characteristics

|   |   | Training | Validation | Test | All |
|---|---|---|---|---|---|
| **N** | Initial | 120 | 251 | 217 | 588 |
|   | Excluded | 0 (0%) | 10 (4%) | 2 (1%) | 12 (2%) |
|   | Measured | 120 (100%) | 241 (96%) | 215 (99%) | 576 (98%) |
| Male |   | 70 (58%) | 126 (52%) | 114 (53%) | 310 (54%) |
| HIV + |   | 65 (54%) | 122 (51%) | 87 (40%) | 274 (48%) |
|   | CD4 count* (cells/µL) | 135 (3–1454) | 213 (1–902) | 396 (3–1353) | 250 (1–1454) |
| Age (years) |   | 34 (18–72) | 36 (18–80) | 37 (18–80) | 36 (18–80) |
| Country | South Africa | 54 (45%) | 134 (56%) | 106 (49%) | 294 (51%) |
|   | Peru | 37 (31%) | 54 (22%) | 98 (46%) | 189 (33%) |
|   | Vietnam | 18 (15%) | 33 (14%) | 11 (5%) | 62 (11%) |
|   | Cambodia | 11 (9%) | 20 (8%) | 0 (0%) | 31 (5%) |
| TB |   | 75 (62%) | 121 (50%) | 56 (26%) | 252 (44%) |
|   | S + C + | 55 (73%) | 101 (83%) | 0 (0%) | 156 (62%) |
|   | S–C + | 20 (27%) | 20 (17%) | 39 (70%) | 79 (31%) |
|   | Clinically diagnosed | 0 (0%) | 0 (0%) | 17 (30%) | 17 (7%) |
| Non-TB |   | 45 (38%) | 120 (50%) | 159 (74%) | 324 (56%) |
|   | Latent TB** | 0 (0%) | 0 (0%) | 89 (56%) | 89 (27%) |

Characteristics of the three cohorts. Values are given as N (%) or median (range).
*CD4 counts were not available for 31 individuals with HIV: 2, 23, and 6 from the training, validation, and test cohorts, respectively. ** Latent TB results were not available for nine individuals in the test cohort.

to demonstrate 90% sensitivity with 5.2% margin of error[22]; (3) an added 40 smear-negative samples distributed between the above two cohorts, which otherwise had none; and (4) a blinded test cohort of 244 samples, limited by the number of smear-negative and culture-negative samples available. (The totals in Table 1 differ because they exclude cases where multiple aliquots of the same sample were shipped by FIND in different cohorts. Results from these duplicate aliquots were combined and assigned to the earliest cohort in which the sample appeared, in order to preserve blinding).

Samples were labeled only with barcodes that contained no clinical information. Investigators had access to the clinical data for the first three cohorts but avoided looking at it until the appropriate stage of data analysis after conducting Simoa assays. Investigators knew the numbers of samples in each TB and HIV category in the blinded test cohort, but nothing about the individual samples, and neither the barcodes nor the order nor any other features of the sample tubes suggested the eventual results. Simoa and AlereLAM assays were conducted independently on the blinded test cohort; a model trained on the prior cohorts was used to predict diagnoses in the test cohort using the Simoa results without seeing the AlereLAM results or any clinical information. Finally, the Simoa concentrations, model predictions, and AlereLAM results in the test cohort were submitted to FIND before FIND returned the clinical results and other information about this cohort.

### Single-molecule array assays
Simoa assays were performed using the Simoa HD-X Analyzer (Quanterix, Billerica, MA). Simoa is based on sandwich ELISAs, which use the selectivity

of a pair of antibodies to quantify an analyte macromolecule. Bead conjugation, detector biotinylation, and Simoa assays were performed according to modified versions of protocols published previously[19]. For the complete detailed protocol, please see the Methods section in the SI.

**Sample preparation.** Frozen urine samples were thawed in a room temperature water bath, inverted to resuspend sediment, and pipette mixed with equal volumes of sample diluent in protein low-binding tubes (Eppendorf). The sample diluent consisted of 2× PBS (pH 7.4, 4 mM phosphate ion, Gibco) with 4% BSA (heat shock fraction, MilliporeSigma), 10 mM ethylenediaminetetraacetic acid (EDTA) (Thermo Fisher), and 0.06% ProClin 300 (MilliporeSigma). Either 210 µL (for one replicate) or 390 µL (for two replicates) of diluted sample was transferred to a 96-well plate (Quanterix) and loaded into the HD-X. At least three replicates were measured for all samples. A calibration curve, consisting of serial dilutions of purified native LAM and Ag85B (BEI Resources) in calibrator diluent (1× PBS with 2% BSA, 5 mM EDTA, and 0.03% ProClin 300), was measured with each batch of samples. The concentrations used for generating the calibration curves were between 0.00256 pg/mL and 8 pg/mL for Ag85B and between 0.256 pg/mL and 800 pg/mL for LAM.

**Data processing.** Concentrations were calculated using calculated $f_{on}$, $I_{bead}$, digital AEB, and analog AEB in the run histories produced by the HD-X software. Analog AEBs for FIND28 (700 nm channel) were calculated according to Zhang et al.[23], with an adjustment to the calculated mean fluorescence intensity of wells with single enzyme molecules, because the background fraction on (approximately 30%) was too high for the HD-X software to calculate it. Replicate AEBs were calculated as a weighted average of digital and analog AEBs as in Zhang et al.[23]. Calibration curves were created by least-squares linear regression of AEB against the concentrations of known calibrators of Ag85B and LAM for each HD-X run. Using the appropriate calibration curves, each replicate of each sample was assigned an Ag85B concentration and three LAM concentrations, one for each capture antibody. The concentration for each sample in each plex was the median concentration of all replicates.

**Assay development and validation.** Simoa assays for 11 *M. tb* antigens in urine (Fig. S1, Tables S1–S3, SI) were validated through dilution linearity, spike recovery, and dropout tests. For the complete procedure, see the Methods section in the SI. The limit of detection (LOD) for each assay was calculated as the concentration equivalent to 3 standard deviations above the background, averaged across batches.

### AlereLAM assay
**Sample and test preparation.** Lateral flow test strips (Determine TB LAM Ag, Abbott 7D2741) were prepared by individually separating each test strip and removing the cover according to the manufacturer's instructions. A total of 244 blinded human urine samples, previously stored at −80 °C, were thawed at room temperature. 80 µL of each sample was centrifuged at 2000 × g for 10 minutes at 4 °C. For each sample, 60 µL of the supernatant was pipetted from the tube and applied to the sample pad on the corresponding test strip. The test strips were allowed to sit in a dry, room temperature environment for 25 minutes before interpreting the results.

**Interpretation of results.** Samples were classified as LAM negative if the "control" line on the test strip was present, but no visible line appeared in the "patient" section. Conversely, samples were considered LAM positive if the "control" line and the "patient" line both were visible. In cases where the "control" line was present and the "patient" line was exceptionally faint, the results were noted but treated as positive cases.

### Statistics and reproducibility
**Replication.** Three replicates were measured from one urine sample for each individual. When Simoa image analysis returned an error (as in 5% of samples) or the three replicates had coefficients of variation above 20%

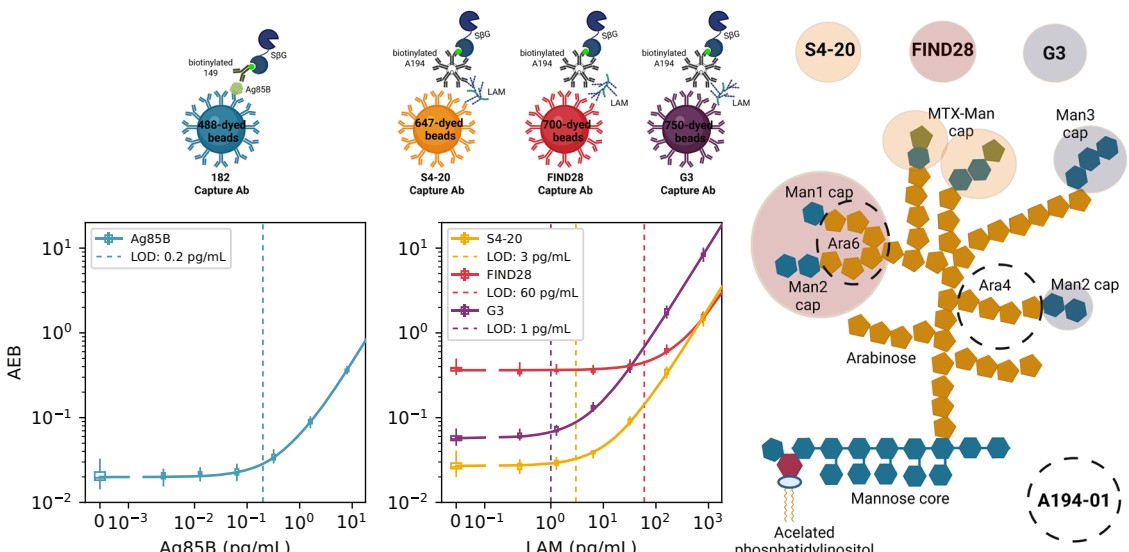

**Fig. 1 | Multiplexed Simoa assays to quantify Ag85B and LAM in urine.** For the Ag85B assay, 182λ-antibody-coated 488-dye-encoded magnetic beads capture Ag85B in urine. For the LAM assays, S4-20, FIND28, and G3 antibodies are coated on 647, 700, and 750 dye-encoded beads and capture LAM in urine. Biotinylated 149 IgG antibodies and A194-01 IgM antibodies bind to Ag85B and LAM, forming sandwich assays. Streptavidin-β-galactosidase (SβG) binds the biotinylated antibodies and transforms its substrate, Resorufin-β-d-galactopyranoside (RGP), to its fluorescent form, which is detected by the HD-X instrument. A camera images and counts the number of fluorescent wells ("on" beads) (meaning they have a protein molecule bound) and the total number of wells containing a bead. The ratio of these values (average enzyme per bead; AEB) is converted back to a concentration according to the corresponding calibration curve. Representative calibration curves are plotted with points (boxplots) summarizing 33 (blank), 11 (third and fifth points), or 12 (all other concentrations) replicates.

(13% of samples), additional replicates were run. The reported concentrations for each sample are the medians of all replicates measured. Twelve samples were excluded because they repeatedly failed Simoa runs; most of these had grossly visible anomalies such as solid masses in the urine. No other samples were excluded; no outliers were removed or adjusted.

**Machine learning.** Cross-validation within the model-building cohort (139 samples) was used to select the model and narrow hyperparameters. Receiver operating characteristic area under the curve (ROC-AUC) was used as the primary performance metric, but balanced accuracy, F-score, sensitivity at ≤1 false positive, and specificity at ≥90% sensitivity were also considered. Hyperparameters were tuned using repeated stratified 5- or 10-fold cross-validation; results from other numbers of folds, ranging from 2 to 55, were used for comparison. Based on performance in the model-building and validation cohorts, a generalized additive model (GAM) of splines was selected after comparing the performance of logistic regression, random forest, and gradient boosting classifiers. For the spline, concentrations were transformed according to

$$X_t = \operatorname{arcsinh}(X/x_0) = \log\left[X/x_0 + \sqrt{(X/x_0)^2 + 1}\right],$$ where $x_0$ is 1/10

the LOD. This resulted in approximately normal features. Feature selection was performed by comparing performance using all possible subsets of the four biomarker levels; use of all four biomarkers was chosen based on noninferiority, though in some cases, in practice, the regularization led one or more features to have no influence.

**Blinded test set.** Hyperparameters (regularization strength, number of knots, and monotonic constraints for each feature) were chosen based on cross-validation within the training and validation cohorts. The final model is available online; see "Data and materials availability" below. The GAM returns a continuous score between 0 and 1, so based on cross-validation within the training and validation cohorts, scores above 0.72 were considered positive, indicating a TB diagnosis according to the model. The model and the corresponding predictions for the blinded test

set were locked on November 4, 2024; on this date, the researchers submitted the Simoa concentrations and predictions to representatives of the FIND specimen bank. The next day, November 5, 2024, FIND provided the researchers with the unblinded clinical results for this cohort, and the researchers proceeded to evaluate the model's performance. No modifications, additions, or exclusions were made to the test data set from the point at which the model was locked down, and neither the test set nor any subset of it had ever been used to assess or refine the model being tested. For 181/215 samples in the test set (84%), all replicates yielded the same predicted diagnosis. Of the 34 (16%) with discordant replicates, the model using the median was correct in 22 (65%), as opposed to 139/181 (77%) of the ones with concordant replicates.

## Results

### A multiplexed assay for Ag85B and multiple forms of LAM in urine

We developed a multiplexed Simoa assay to quantify Ag85B and LAM in urine. These analytes were chosen from a panel of 11 *M. tb* antigens for which Simoa assays were developed and tested in urine from individuals with and without TB (Tables S1–S3 and Fig. S1, SI. Ag85B and LAM were the only biomarkers detected in over half of the TB-positive samples and exhibited a statistically significant difference between TB and non-TB samples in the discovery cohort. The assay format is depicted in Fig. 1. Ag85B is measured with a pair of monoclonal antibodies at a LOD of 0.2 pg/mL; LAM is captured by three different monoclonal antibodies that bind to distinctly different epitopes: S4-20 (MTX-Man caps), FIND28 (Ara6 with or without Man caps), and G3 (Man caps without MTX) at LODs of 3, 60, and 1 pg/mL respectively. A single detection antibody, A194-01 IgM, binds to Ara4 and Ara6 moieties with or without Man1 and MTX caps (Fig. 1). These antibody pairs for LAM were chosen after extensive cross-testing of available antibodies using pooled urine from individuals with and without TB (Fig. S2, SI), combined with prior findings from the literature[11,13,20]. They include the antibody pairs used by FujiLAM and MSD (S4-20/A194-01) and the Simoa LAM serum assay from Brock et al. (FIND28/A194-01)[11,20], but the diagnostic measurement of different LAM channels plus Ag85B is unique to this work. Concentrations are determined

from the fluorescent signal readout using linear calibration curves, so each sample is given one concentration for Ag85B and three concentrations for LAM, one for each capture antibody clone.

Urine samples are diluted twofold, so the analytical limits of detection of 0.2, 3, 60, and 1 pg/mL correspond to 0.4, 6, 120, and 2 pg/mL in urine samples. The assays do not cross-react: the presence of LAM does not affect the measured Ag85B concentration or vice versa (Fig. S3, SI). Urine samples dilute linearly into the sample diluent (parallelism) and also when blended with other urine samples (admixture linearity), suggesting no differences in the detector antibody binding affinity to endogenous analytes and standard analytes and showing the flexibility of the assay at varying dilutions (Figs. S4 and S5, SI). Specificity for native Ag85B was confirmed via immunoprecipitation and silver stain in wild-type and Ag85B deletion mutant BCG (Fig. S6, SI). When known quantities of Ag85B and LAM were spiked into urine samples, the measured LAM concentrations were confirmed to equal the sum of the endogenous and spiked concentrations, with recoveries averaging 107% for S4-20, 103% for FIND28, and 130% for G3. The measured Ag85B concentrations were lower than the sum of the endogenous and spiked concentrations, with an average recovery of 54% (Fig. S7, SI). The recovery of Ag85B was negatively affected by the urea concentration in the urine, especially at low protein concentration samples (Fig. S8, SI). We found that the urea in the urine attenuated the signal of the assay (Fig. S9, SI). Ag85B renaturation attempts using TMAO did not yield higher recoveries (Fig. S10, SI). Adding urea to the calibrators improved the recoveries to some extent (Fig. S10, SI). However, the urea concentration between individuals varies, with concentrations ranging between ~17–166 mM, independent of Ag85B concentrations (Fig. S9, SI). Urea was not added to the calibration curve because any one concentration would not translate well across different samples.

## Accuracy in diagnosing TB

Using this multiplex assay for Ag85B and three different LAM forms, we measured samples from 576 individuals (Table 1). Each urine sample corresponded to a unique adult with symptoms suggestive of pulmonary TB. All urine samples were collected before treatment. The different TB categories and classification for the samples are depicted in Table S5. All three cohorts were approximately evenly split between men and women and people with and without HIV. The training and validation cohorts were predominantly smear-positive, while the test cohort was exclusively smear-negative (Table 1). The test cohort was fully blinded: the experimenters had access to the numbers of positive and negative samples but not to any information about each sample until after the model was evaluated.

Proceeding in multiple stages (Fig. S11, SI), we trained a GAM classifier to combine the concentrations of Ag85B and the three LAM measures into a single diagnostic result[24]. The GAM learns nonlinear relationships between each biomarker concentration and TB status, then combines these relationships to produce a continuous diagnostic score between 0 and 1. This data-driven approach automatically determines how to weight and combine the four biomarkers without requiring pre-determined cutoffs or simple arithmetic combinations. A model was trained on the model-building and validation cohorts (361 samples) and evaluated on the blinded test cohort (215 samples) (Fig. S11, SI; Fig. 2, green line). Finally, to generate the most robust estimates and confidence intervals for accuracy metrics, repeated nested stratified five-fold cross-validation (CV) was used on the entire cohort (576 samples) (Fig. 2, black lines and gray ranges). In this nested cross-validation, 20% of the samples were set aside for testing; cross-validation was used on the remaining 80% to choose the model hyperparameters that best balanced bias and variance; a model was trained on the 80% with the best hyperparameters and tested on the remaining 20%. Repeating this process many times provided an unbiased estimate of the performance of our diagnostic test without data leakage, as well as minimally biased confidence intervals[25,26]. The model performance characteristics in the nested cross-validation are shown in Table 2. The blinded ROC curve aligns with the nested CV curve for smear-negative patients (Fig. 2).

In Fig. 2, the model's performance in various subsets is shown in ROC plots, showing true positive rate (sensitivity) versus false positive rate (1-specificity). Overall, the sensitivity of our assay is 45% (95% CI: 44%–47%) with 98% specificity (95% CI: 97.6%–98.3%) (Table 2). The contribution of having a four-plex assay versus other combinations was also evaluated by calculating the AUC-ROC scores for each individual biomarker (Fig. S13, SI). Adding Ag85B to all other combinations improved the AUC-ROC scores (Fig. S13, SI). Including HIV status, sex, and age as predictors did not improve the performance of the model in initial tests, so they were not used. According to our estimated TB probability, nine out of 17 samples classified as Clinical TB (53%) were detected as positive TB, and 26 out of the 27 samples (96%) classified as "Likely subclinical TB" were not detected as TB.

The results of AlereLAM using the 215 samples from the blinded test cohort were compared with the results of our multiplex assay (Table 3). Overall, the sensitivities of AlereLAM and the multiplex Simoa assay were 12.5% (7/56) and 41% (23/56), and the specificities were 93% (148/159) and 87.4% (139/159). Similar results for the AlereLAM test were seen using 29 additional samples from the training and validation cohorts (Table S6). Because AlereLAM is approved for HIV-positive patients only, we also stratified the samples by HIV status. For HIV-positive patients, the sensitivities of AlereLAM and the multiplex Simoa assay were 23.8% (5/21) and 28.6% (6/21), and the specificities were 93.9% (62/66) and 95.5% (63/66). Simoa's sensitivity in this blinded test cohort is much lower than its overall sensitivity because the test cohort consisted exclusively of smear-negative samples.

## Ag85B and LAM concentrations in urine: effects of country of origin, HIV status, and antigen heterogeneity

Figure 3 shows boxplots for the concentrations of Ag85B and LAM (according to the three capture antibodies), separated by TB diagnosis, HIV status, and country. The LAM concentration differs for each capture antibody, as it identifies different epitopes on LAM, and their prevalence differs.

Overall, there are significant differences in Ag85B and LAM concentrations between TB and non-TB according to all antibodies, except for G3 in HIV-negative samples. However, Ag85B and LAM concentrations according to S4-20 give the highest separation. Overall, TB + HIV+ individuals have significantly higher concentrations of Ag85B and LAM according to all antibodies than TB + HIV– individuals. The concentrations of Ag85B and LAM in TB + HIV+ individuals from Vietnam are significantly higher than in Peru and South Africa. In contrast, TB + HIV-individuals from Vietnam have significantly lower concentrations of Ag85B and LAM than those from other countries. In Peru, the concentrations of Ag85B and LAM in non-TB and TB individuals overlap in all biomarkers except for S4-20, indicating that Peru has a high rate of false positives. The LAM concentration according to S4-20 contributes the most to the model's decision (Fig. S13). The addition of Ag85B slightly improves the model, and G3 and FIND28 contribute less (Fig. S13).

## Discussion

Although TB is curable using antibiotics, it is still the leading infectious cause of death worldwide. The main challenges related to the diagnosis of TB include the use of biohazardous sputum, insufficient sensitivity, and slow turnaround time for the gold standard culture. Here, we present a multiplex biomarker assay in urine that could be used as an adjunctive test to diagnose TB.

Our test includes the protein biomarker Ag85B and three capture antibodies for the glycolipid biomarker LAM. As these three capture antibodies identify different epitopes of LAM, we had hypothesized that combining the results of all three binding pairs would improve the assay's performance. Concentrations for all three capture antibodies are calculated using the same LAM standard, but they perform quite differently: S4-20 best separates people with and without TB, FIND28 reports the highest concentrations in urine on average, and G3 has the best analytical sensitivity. S4-20 was less susceptible to both background nonspecific binding (Fig. 1) and

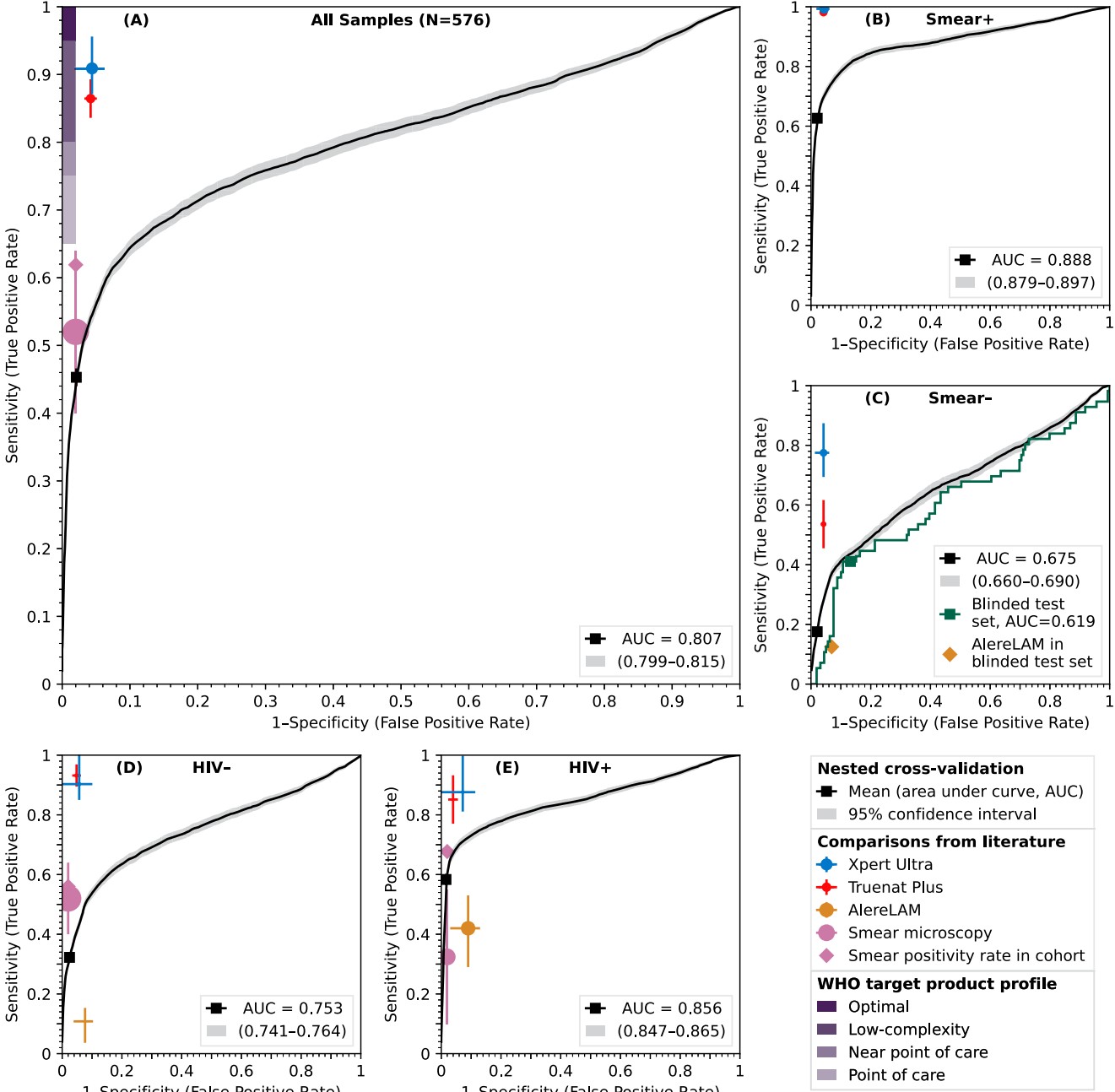

**Fig. 2 | Receiver operating characteristic (ROC) curves demonstrating the model's sensitivity and specificity.** The model's performance is stratified by HIV status and smear result: **A** overall, (**B**) smear-positive TB versus non-TB, (**C**) smear-negative TB versus non-TB, (**D**) people without HIV, (**E**) people with HIV. The black line and associated gray range give the mean and 95% confidence interval across all samples in the cohort, obtained using nested cross-validation (CV). Squares give the empirical sensitivity and specificity at predetermined thresholds. The WHO optimal and minimal ranges are depicted in shaded purple rectangles in the upper left corner. Xpert Ultra, Truenat Plus, AlereLAM, and smear microscopy performances and cross reactivity with components present in the urine of people without TB (Fig. 3), so it is understandable that it contributed most to the model. These results are consistent with previous reports[11]. Potential cross-reactivity of the different capture antibodies with other glycolipids or lipopolysaccharides present in urine may explain the variation in detected LAM concentrations. For example, Sigal et al. reported cross-reactivity of different antibody pairs with other *Mycobacterium* species and non-Mycobacteria. Specifically, the FIND28/A194-01 pair had a poor specificity of 63% in urine, with some TB-

confidence intervals are depicted by blue, red, gold, and salmon crosses, respectively, with larger circles corresponding to larger cohorts. The actual rates of smear positivity in the cohorts are indicated with salmon diamonds. **C** The model's performance in the blinded test set after training on the training and validation cohorts is shown in green, and performs similarly to the results from nested CV. The green square indicates sensitivity and specificity at predetermined thresholds, and the gold diamond gives the empirical performance of AlereLAM in the blinded test set. A summary of the characteristics of the common TB diagnostic tests used for comparison in this figure can be found in Table S4.

negative samples giving signals as high as 10-fold above the blank signal. The same pair was shown to cross-react with the non-mycobacterial actino-mycetes *Nocardia*, *Gordonia*, *Rhodococcus*, and *Tsukamurella*, as well as cross-react with the mycobacteria *M. fortuitum* and *M. smegmatis*. In contrast, the S4-20/A194-01 pair had no cross reactivity with fast-growing mycobacteria and non-mycobacterial actinomycetes, evident by its higher specificity of 97%[11]. The higher concentrations of Ag85B and LAM in TB + HIV+ than TB + HIV− are probably due to the suppressed immune

**Table 2 | Model performance characteristics in nested cross-validation**

| | | Sensitivity | Specificity | ROC-AUC |
|---|---|---|---|---|
| All (N = 576) | Overall | 0.45 | 0.979 | 0.81 |
| | 95% CI | (0.44–0.47) | (0.976–0.983) | (0.80–0.81) |
| | Smear + | 0.63 | | |
| | 95% CI | (0.61–0.64) | | |
| | Smear– | 0.18 | | |
| | 95% CI | (0.16–0.19) | | |
| HIV– (N = 302) | Overall | 0.32 | 0.975 | 0.75 |
| | 95% CI | (0.31–0.34) | (0.970–0.981) | (0.74–0.76) |
| | Smear + | 0.48 | | |
| | 95% CI | (0.45–0.50) | | |
| | Smear– | 0.13 | | |
| | 95% CI | (0.11–0.15) | | |
| HIV + (N = 274) | Overall | 0.58 | 0.984 | 0.86 |
| | 95% CI | (0.56–0.60) | (0.980–0.988) | (0.85–0.87) |
| | Smear + | 0.75 | | |
| | 95% CI | (0.73–0.77) | | |
| | Smear– | 0.24 | | |
| | 95% CI | (0.21–0.28) | | |

**Table 3 | Comparison of AlereLAM results and Simoa multiplex assay**

| | | AlereLAM | Multiplex Simoa assay | FIND |
|---|---|---|---|---|
| All (N = 215) | Positive | 18 | 43 | 56 |
| | Negative | 197 | 172 | 159 |
| | TP | 7 | 23 | |
| | FN | 49 | 33 | |
| | TN | 148 | 139 | |
| | FP | 11 | 20 | |
| | Sensitivity | 12.5% | 41% | |
| | Specificity | 93% | 87.4% | |
| HIV + (N = 87) | Positive | 9 | 9 | 21 |
| | Negative | 78 | 78 | 66 |
| | TP | 5 | 6 | |
| | FN | 16 | 15 | |
| | TN | 62 | 63 | |
| | FP | 4 | 3 | |
| | Sensitivity | 23.8% | 28.6% | |
| | Specificity | 93.9% | 95.5% | |

*TP* true positive, *FN* false negative, *TN* true negative, *FP* false positive.

system in individuals living with HIV, leading to higher bacterial load and more of these biomarkers in the bloodstream and urine[27]. Late-stage HIV infection can impair kidney function and also lead to increased filtration of LAM into the urine[28]. Another possibility is that HIV-positive patients can often develop disseminated TB, including infection of the kidneys, which allows LAM to enter the urine directly from the infected tissue[29].

The 2024 edition of the TPP did not relax the 2014 edition's criteria for sensitivity or specificity[8,9]. But in the intervening decade, diagnostics have been introduced that fall below its stringent minimum specificity (98%) relative to culture: Xpert Ultra (96%), AlereLAM (91%), and Truenat Plus (96%)[7,30–32]. This reflects the importance and difficulty of detecting TB, as well as the limitations of culture as a gold standard: some "false positives" according to index assays may well be false negatives of the reference assay[33]. However, although our test is less sensitive than Xpert Ultra and Truenat Plus, it uses safe and accessible urine samples, rather than sputum. Our assay is a step towards a diagnostic test that may supplement existing diagnostics and aid clinicians in deciding whether to start TB treatment. For example, the AlereLAM assay is recommended for HIV-positive patients, although its sensitivity is quite low (42% sensitivity at 91% specificity)[7]. Because our test has 58% sensitivity at 98% specificity among people with HIV, it can serve as a better diagnostic test for this population. On the other hand, because our assay has a sensitivity of 63% among smear-positive individuals at a specificity of 98%, it is not ready to replace smear microscopy.

If this study is generalizable, our assay has diagnostic accuracy (sensitivity/specificity) modestly superior to AlereLAM; inferior to culture, Xpert Ultra, and Truenat Plus; and overlapping with smear microscopy (Fig. 2). It would be comparable to an updated FujiLAM assay (2024, called FujiLAM II or FujiLAM2): a retrospective study demonstrated 80% sensitivity (55/69 samples) and 93% specificity (80/86 samples) in people with HIV, while another paper reported lower sensitivity in TB meningitis[18,34,35]. In addition, it is more sensitive but slightly less specific than the serum LAM Simoa assay developed by Brock et al.[20]; less sensitive in HIV-negative people than the MSD EclLAM assay developed by Sigal et al.[36]; and more specific than the Simoa cytokine triage test developed by Ahmad et al.[21]. (Neither the new FujiLAM studies nor Ahmad et al. reported smear positivity rates, which are a major determinant of ease of diagnosis.)

This study has several limitations. It employed a retrospective case-control study design, preventing the determination of positive and negative predictive values in a real-life context. Such a design also risks over-estimating performance due to the potentially biased inclusion of patients at extremes of phenotype distribution that might not represent the target population (spectrum bias)[33]. We attempted to mitigate this by including individuals classified as "clinically diagnosed TB," and all individuals (cases and controls) had TB-like symptoms. Samples were chosen based on target numbers of HIV, culture, smear, and latent TB status, introducing a selection bias. Despite efforts to stratify, our cohort may not be representative of all people who present for TB testing at a given center, or of the combined populations in the meta-analyses for other tests. Because our cohort was chosen to include a certain percentage of smear-positive and smear-negative samples, a direct comparison with the literature-reported performance for smear microscopy is problematic. (Because AlereLAM in the blinded test cohort was conducted alongside but independently of Simoa, it is a more appropriate comparator, subject only to the biases inherent in the study design.) Case-control diagnostic studies are considered less generalizable than prospective studies, but as long as data collection is planned before testing and samples represent the full spectrum of the disease, they have not been found to have a consistent bias relative to prospective studies[37].

Another limitation is the age of the samples used. Out of 576 samples, 77 were frozen from 2012–2015, and 511 were frozen from 2016–2019. Using fresh samples or samples frozen for a shorter period in a prospective study might improve assay performance.

We performed a blinded validation to evaluate the model's performance. However, the TB samples from the test cohort were all smear-negative, so we did not have a truly blinded validation of our assay's performance in roughly half of the people with TB who are smear-positive. The strong correspondence between results in the blinded test cohort and nested cross-validation (green and black lines, respectively, in Fig. 2) is consistent with theoretical and simulated findings that nested cross-validation is an unbiased estimator of model performance[26]. Further independent validation in a prospective, real-world setting is warranted to assess the true performance of this diagnostic test.

Significant heterogeneity was observed in biomarker concentrations across countries (Fig. 3). Although we cannot explicitly determine the origin of this heterogeneity, we suggest several possibilities. One possible explanation is the different times of sampling across countries. In some countries,

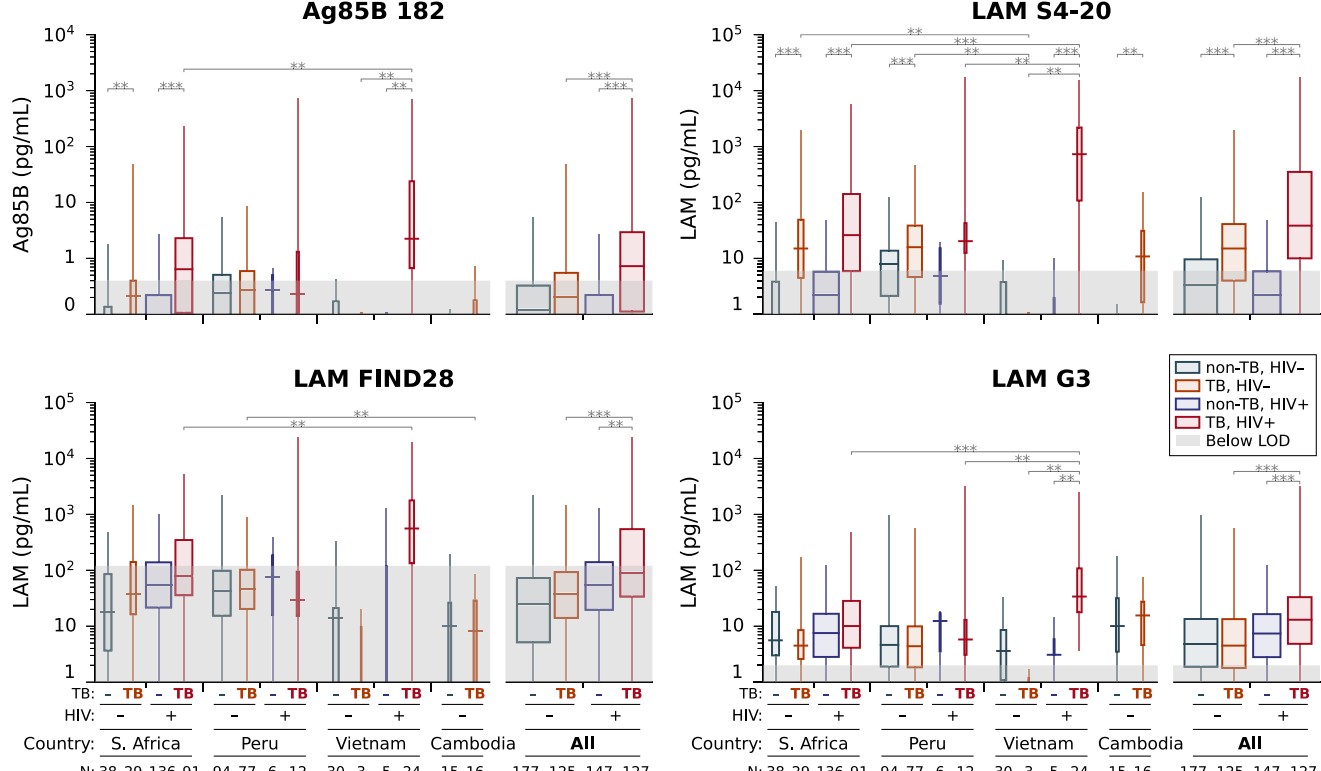

**Fig. 3 | Box plots of Ag85B and LAM concentrations in urine.** For each of the four assays (Ag85B and three LAM capture antibodies), results are separated by TB diagnosis, country, and HIV status. Boxes cover interquartile ranges, with medians as lines inside, and whiskers extending to the full range. Groups were compared using the Mann–Whitney $U$ test (two-sided between countries, one-sided within countries, with Benjamini-Hochberg correction for multiple comparisons): **$p < 0.01$, ***$p < 0.001$. Measurements below the LOD for each assay are indicated by a shadowed box. The width of each box is proportional to the number of independent individual samples in that group.

there are extreme delays in patients seeking treatment, which is also one of the major hurdles to controlling TB. For example, a four-week delay in seeking treatment was seen in KwaZulu Natal, South Africa, and a ten-week delay in South Africa's rural Northern Province, now officially known as Limpopo, which far exceeds the WHO-recommended two weeks for initiating treatment after suspicion[38]. Reasons for delay include distance from diagnostic facilities, long waiting times, and absence of clinical symptoms[38,39]. Delays in diagnostic testing in different countries can be part of the cause for lower concentrations of biomarkers in the urine at the time of sampling. It may be valuable to record the number of days from the onset of symptoms to seeking a diagnosis, especially when acquiring samples for clinical trials, as the concentration of biomarkers can be affected. (FIND collected a "binned" version of symptom duration for some individuals in our study, but the data were too coarse for this type of analysis.) In addition, this information may aid in assessing pre-test probabilities.

Another explanation for the heterogeneity in biomarker concentrations across countries may be attributed to different LAM compositions in different lineages. *M. tb* has seven lineages: in South Africa and Peru, the dominant lineage is lineage 4, in Vietnam lineages 1 and 2[40,41], and in Cambodia lineage 1[42]. There is evidence that the relative mannose capping of LAM differs across lineages[43]. The difference in LAM capping can result in different binding of the antibodies to LAM, and therefore, affect the measured concentrations in urine. The overlap between the concentrations of Ag85B and LAM in non-TB and TB individuals in Peru could indicate that in Peru, other mycobacteria are present to which the antibodies bind. In future studies, such as the prospective validation of the present assay, it would be instructive to conduct mycobacterial sequencing and compare biomarker abundance results with lineage and phenotype information.

The HD-X instrument used in this study would need to be simplified in order to become suitable for operating in low and middle income countries (LMIC). For example, the capital cost and instrumentation size could be

reduced by porting the assay to a benchtop Simoa system like the Quanterix SR-X, or by converting the assay to MOSAIC, which uses a flow cytometer readout[44,45]. Alternatively, the assay could be adapted to a readout device with fewer multiplex channels by narrowing the number of biomarkers and affinity agents (Fig. S13). In this case, simpler flow cytometers, like the one described by Cheng et al.[46], could be used.

In conclusion, we have demonstrated that different antibody pairs report diverging LAM concentrations, indicating that "LAM concentration" is not a well-defined assay-independent quantity. Nevertheless, we did not find that measuring LAM with multiple antibodies improved clinical performance. Despite our significant advances in analytical sensitivity, LAM remains below the detection limit of an ultrasensitive assay, which may be the reason why our diagnostic achieved only modest clinical sensitivity. We hope that the present test can be translated into a clinical laboratory assay using MOSAIC to conduct readout with a flow cytometer, and that its large set of LAM and Ag85B measurements can serve as the groundwork for the development of other diagnostics for TB.

## Data availability
All data are available on GitHub at https://github.com/tylerdougan/simoa-tb and archived on Zenodo at https://doi.org/10.5281/zenodo.17867202[47]. This repository includes directories for input (AlereLAM readings, assay validation Simoa results, external files of de-identified clinical metadata from FIND and meta-analytic tabulation of comparison assays' performances, and Simoa sample results); output (including CSV data for each figure, prefaced with the figure number); and processed intermediate data.

## Code availability
All code used to conduct the analysis and generate the figures and tables is available on GitHub at https://github.com/tylerdougan/simoa-tb and archived on Zenodo at https://doi.org/10.5281/zenodo.17867202 (the same

repositories as the data)[47]. The source code contains Jupyter notebooks and auxiliary Python scripts.

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

## Acknowledgements

Funding was provided through Gates Foundation grant INV-009043 (formerly OPP1157033) to DRW. We thank Bruce P. Bausk and Rabsa Sikder for conducting experiments that led to this study, as well as Ming Yang Lu and Faisal Mahmood of Brigham and Women's Hospital for data analysis of the same. We thank Emmanuel Moreau of FIND for organizing and providing urine samples, Abraham Pinter of Rutgers University, and Masanori Kawasaki and Ryo Higashiyama of Otsuka Pharmaceutical Co. for kindly providing antibodies to LAM. We are grateful to Joel Ernst of UCSF for providing BCG cell lines, and Bryan Bryson of MIT for lysing them. We thank Olga Demler and Sebastian Cajas for their helpful suggestions on machine learning. Finally, we are grateful to the study participants who graciously offered biological samples and access to aspects of their clinical records. Figure 1 was created in https://BioRender.com.

## Author contributions

Conceptualization: D.R.W, L.X; Methodology: L.X, T.J.D, S.R; Software: T.J.D; Validation: T.J.D, S.R; Formal analysis: T.J.D, S.R; Investigation: T.J.D, S.R, S.C.D, L.X; Resources: D.R.W, T.J.D; Data curation: T.J.D; Writing—original draft: T.J.D, S.R, L.X; Writing—review & editing: D.R.W, S.R, T.J.D, S.C.D; Visualization: T.J.D, S.R; Supervision: D.R.W; Project administration: D.R.W; Funding acquisition: D.R.W, L.X.

## Competing interests

The authors declare the following competing financial interest(s): David R. Walt is a founder, equity holder, and Director of Quanterix Corporation. Dr. Walt's interests were reviewed and are managed by Brigham and Women's Hospital and Mass General Brigham in accordance with their conflict-of-interest policies. The remaining authors declare no competing interests.
