## [Transparent Peer Review file · Communications Medicine]

Antigen heterogeneity in the development and clinical validation of a multiplexed urine test for tuberculosis

Corresponding Author: Professor David Walt

Version 0:

Reviewer comments:

Reviewer #1

(Remarks to the Author)

The authors describe the development of a new TB diagnostic test in urine samples by combining the detection of LAM and Ag85C using four different monoclonal antibodies (3 for LAM and 1 for Ag85B). The authors use single molecule array (Simoa) method for detection of TB antigens in urine samples. The assay was validated on case and control samples from 550 individuals. Previous studies have reported the use of Simoa for detect LAM in serum from TB patients and AlereLAM is a lateral-flow immunochromatographic assay that detects LAM using polyclonal antibodies in urine and is approved for TB diagnostics in HIV-positive individuals. The novelty of this study is in multiplexing detection of LAM and Ag85B together with use of a more sensitive single molecule-based detection technique. The assay has an overall sensitivity of 45% and specificity of 98% but performs worst in smear-negative test cohort. The assays overall sensitivity is too low to be used as a confirmatory TB diagnostic test. The assay yields overall better sensitivity (58%) in HIV-positive individuals when compared to AlereLAM but less sensitive than EclLAM in HIV negative individuals. In handful of samples the test detected 50% of clinical TB cases as positive and 96% likely subclinical TB cases as negative. Authors propose the new assay as a replacement for AlereLAM.

The work's primary interest is in showing that adding additional antibody pairs for LAM does not substantially improve test accuracy, supporting the notion that LAM in the urine of many TB patients may be below the level of detection of a highly sensitive assay, and that further work along these lines may not demonstrate substantial improvements in sensitivity.

Introduction –

- Truenat is also WHO approved (as it reads suggests that only Xpert is approved)
- Shorten the introduction. For example, remove the “In summary” paragraph. Shorten the description of Simoa assay and use references instead. Also remove the discussion of findings from the final paragraph of the introduction.
- Remove the word ultrasensitive to describe the test.
- Replace the word authoritative with confirmatory or similar.

Methods –

- Include how the LOD was calculated.
- In methods section include the range of LAM and Ag85B concentrations used for generating the calibration curves.
- The paragraph describing selection of cohorts includes different numbers for training/validation/test in different parts of the paragraph, eg, blinded test cohort is variably described as 217 or 244 participants. This is explained in Figure S9, where again this number is different (215 samples) but could be clearer in the text. In the results section the number for this cohort is 215 samples.
- Where repeat testing of the different aliquots of the same specimen was done (as described in methods), were results always concordant?
- Variation in replicate testing for each sample should be described in more detail – for what proportion of samples did additional replicates need to be run because CVs were >20%? In how many cases were replicates distributed across the positive/negative threshold?
- I found Figure S9 hard to reconcile with the explanatory test under subheading 2. It appears that the final analysis included in Figure 2 (black model) was trained completely separately from the initial training model (green), and was done unblinded.

Presumably this repeated training was needed because the green model had been trained on largely smear positive samples while the test set was smear negative samples. This seems suboptimal – it might have been better to train the model on a mix of smear/pos and neg and then test a similar ‘test’ set in an blinded manner.

Results-

- The rationale for choosing Ag85B and LAM from the 11 Mtb antigens tested in simoa assay (Table S3) in urine is unclear.
- Could cross reactivity with LAM-like proteins explain the variation between S4, FIND and G3 in detected LAM concentrations?
- Fig S8, S1 is referenced in the wrong statement. Move this to one above.
- Figure 1 add x-axis labels
- Subhead 2: Give full form of ROC and CV upon first use.
- Figure 2 legend: “shaded purple rectangles in the upper left-hand corner” not right-hand corner.
- Figure 3: Unclear how there are values of LAM and Ag85B in urine that are below the LOD for each antibody? Are these log values – if so, please relabel axes. Add median and interquartile ranges to the dot plots.
- Figure S13: What do the values mean? Label for capture and detection antibody. If numbers refer to compatibility of capture/detection antibodies, not sure why if FIND28 has not worked with A194-01 detection antibody, why this combination was chosen?
- What is meant by C in HIV+C+ and HIV-C+ samples (results)? Presumably culture, but this is not spelt out anywhere.
- The increment in sensitivity of Simoa over Alere seem to be primarily in HIV- patients, whilst sensitivity was similar in HIV-positive patients. Do the authors have a possible explanation for this?

Discussion-

- “FIND28 measures the highest concentrations in urine on average” – is this actual detection of LAM or simply non-specific background, given that the distribution of concentrations is almost the same in TB and non-TB patients (Figure 3)? What is the specificity of each of these monoclonal antibodies for detection of antigen in urine? Include in discussion if known from previous studies.
- This statement at the end of the results section: “For HIV-positive patients, the sensitivities of AlereLAM and the multiplex Simoa assay were 23.8% (5/21) and 28.6% (6/21)” – I thought sensitivity in HIV+ was 68% for Simoa? Perhaps this reflects only smear negative samples (hard to tell from paragraph construction)?
- For costing, I suspect that the \$6 quoted per replicate reflects actual reagent costs and not the true cost that the assay would be offered at commercially, given its complexity and need for a margin. A more realistic costing might be useful.
- The test format seems relatively complex and there is a need for complex instrumentation. It seems unlikely that, in this format, the test could be implemented at or close to POC in LMIC.
- I was confused by this statement: “In addition, the ability to detect 96% of the “likely subclinical” samples as negatives may suggest that our assay can be used in the future as a confirmatory test for this population.” Presumably in this population you want to identify these patients as positives, rather than negatives, as they would benefit from treatment. Are you assuming that the majority of those identified as ‘likely subclinical’ did not in fact have subclinical TB (i.e., were true negatives)?
- The authors do not provide a rationale for the use of the multiplex assay as there appears to be no added advantage of using the multiplex compared to S4-20 alone based on AUC scores.

Reviewer #2

(Remarks to the Author)

I read with interest study by anonymous authors entitled “Development and validation of a multiplexed single-molecule array urine test for tuberculosis: A case-control diagnostic accuracy study”.

Subject of the study is clearly of interest and worth exploring given a recent rise in TB cases in many settings and well known challenges associated with TB disease diagnosis especially in resource constrained settings, where lack of access to healthcare and timely diagnosis is exacerbated by intrinsic TB laboratory diagnosis problems in those living with HIV. While the study is generally technologically sound, I think it has quite a few weaknesses on medical and programmatic sides highlighted below. I also felt that study will benefit from re-structuring and significant shortening as follows:

- There are no page and/or line numbers which makes reviewing the paper and making any comments extremely challenging.
- Introduction sections spans over 6 pages which is way too long. Will benefit from substantial editing to remove unnecessary details (like performance of different diagnostic assays etc) leaving up to 2 pages max;
- I am not clear on how specificity was calculated in different cohorts. Some cohorts included “clinical TB” specimens – have these been included in specificity and/or sensitivity calculations? What about latent TB specimens? Figure 2 does not contain these subsets and while some results are provided, I would like to see these in one of tables with clearer explanations.
- I am not sure what section “Ag85B and LAM concentrations in urine” adds and how this could be interpreted. Ultimately Figures 3 and 4 show that concentrations of all analytes tested (Ag85B and three LAMs) overlap significantly in TB vs non-TB samples, and also vary significantly across settings. Why concentrations of analytes are so different (up to statistical significance) across different settings? How would this affect assay performance in different settings? Does it mean that assay may work, for instance, in Vietnam, but less likely so in Peru?

- I cannot fully understand last paragraph of Results section. Authors tested their assay in parallel with commercial Alere LAM assay on two subsets of samples comprising 215 and 244 specimens. What are these samples? Why sensitivity of multiplex Simoa in samples collected from PLHIV was significantly lower in these experiments compared to claimed sensitivity in overall cohort (28.6% in a head to head comparison vs 58% in training and validation cohorts)?

- What are the interpretation rules of the multiplex assay? I mean, how are Ag58B quantification results combined with LAM detection results?

- Authors claim that their assay is more sensitive in PLHIV compared to the only commercially available assay Alere LAM. Firstly, I would like to see more recent references, for example Adzemovic et al., CID 2025 where results of a latest version of Fujifilm TB LAM assay evaluation in Uganda were published; assay has not yet been endorsed by WHO but it's claimed to employ an ultrasensitive LAM detection technology so it's worth including in discussion. Secondly, I would like to see considerations regarding potential commercialization of Simoa assays in terms of technology and prices – at the moment Simoa seems too difficult to commercialise which should be considered in Discussion section to put in the context of unmet medical needs, cost effectiveness and health technology assessment.

Reviewer #3

(Remarks to the Author)

I co-reviewed this manuscript with one of the reviewers who provided the listed reports. This is part of the Communications Medicine initiative to facilitate training in peer review and to provide appropriate recognition for Early Career Researchers who co-review manuscripts.

Version 1:

Reviewer comments:

Reviewer #1

(Remarks to the Author)

The authors have responded comprehensively to our comments and suggestions, and the paper reads more clearly now. We have no further comments.

Reviewer #2

(Remarks to the Author)

I would like to thank authors for addressing my and other reviewers' comments. While the manuscript has benefited from the modifications, I still think there is a room for improvements and further clarifications as follows:

- Introduction section is still too heavy. Suggest further redacting it and remove or significantly reduce fragments related to LAM (lines 112 -133).

- I am concerned about classifying subclinical TB as negative (i.e. non-TB, line 234) for the purposes of the assay validation. Author mention that those with subclinical TB had changes on their CXR suggestive of TB and also had TB cultures on subsequent visits. This essentially means that those with subclinical TB had an active TB (vs latent TB infection) and SHOULD be detected using rapid assays including any LAM- or Ag85B based tests.

Author report that 26/27 (96%) samples from patients with subclinical TB tested negative on their assay translating into sensitivity for subclinical TB of just 3.7%. This require explanations that need to be added to Discussion section.

- I am also not convinced with explanations on why performance of the assay differs across settings. Prevalence of different genetic families/lineages so differ across settings but chemical composition of LAM/Ag85B antigens do not. Please try to provide some evidence on why lineages may be associated with varying concentrations of antigens in urine - to be honest, I cannot come up with such an explanation.

Reviewer #3

(Remarks to the Author)

I co-reviewed this manuscript with one of the reviewers who provided the listed reports. This is part of the Communications Medicine initiative to facilitate training in peer review and to provide appropriate recognition for Early Career Researchers who co-review manuscripts.

Version 2:

Reviewer comments:

Reviewer #2

(Remarks to the Author)

Thank you for addressing my last round of comments

Reviewer #1 (Remarks to the Author):

The authors describe the development of a new TB diagnostic test in urine samples by combining the detection of LAM and Ag85C using four different monoclonal antibodies (3 for LAM and 1 for Ag85B). The authors use single molecule array (Simoa) method for detection of TB antigens in urine samples. The assay was validated on case and control samples from 550 individuals. Previous studies have reported the use of Simoa for detect LAM in serum from TB patients and AlereLAM is a lateralflow immunochromatographic assay that detects LAM using polyclonal antibodies in urine and is approved for TB diagnostics in HIV-positive individuals. The novelty of this study is in multiplexing detection of LAM and Ag85B together with use of a more sensitive single molecule-based detection technique. The assay has an overall sensitivity of 45% and specificity of 98% but performs worst in smear-negative test cohort. The assays overall sensitivity is too low to be used as a confirmatory TB diagnostic test. The assay yields overall better sensitivity (58%) in HIV-positive individuals when compared to AlereLAM but less sensitive than EclLAM in HIV negative individuals. In handful of samples the test detected 50% of clinical TB cases as positive and 96% likely subclinical TB cases as negative. Authors propose the new assay as a replacement for AlereLAM

The work's primary interest is in showing that adding additional antibody pairs for LAM does not substantially improve test accuracy, supporting the notion that LAM in the urine of many TB patients may be below the level of detection of a highly sensitive assay, and that further work along these lines may not demonstrate substantial improvements in sensitivity.

Introduction –

- Truenat is also WHO approved (as it reads suggests that only Xpert is approved)*

We appreciate the reviewer for bringing this to our attention. As part of shortening the Introduction (see next comment), we have deleted the paragraph including this information.

- Shorten the introduction. For example, remove the “In summary” paragraph. Shorten the description of Simoa assay and use references instead. Also remove the discussion of findings from the final paragraph of the introduction.*

We have shortened the Introduction according to the suggestions. The description of the Simoa assay has been moved to the SI and the discussion of findings was deleted from the final paragraph of the Introduction. Parts of the “In summary” paragraph have been merged with the paragraph starting with “About 60% of TB

diagnoses are made with culture...” (page 4 of the marked-up file; page 3 of the clean file).

- *Remove the word ultrasensitive to describe the test.*

Removed.

- *Replace the word authoritative with confirmatory or similar.*

We appreciate the highlight of this wording issue. We have revised the sentence to state, “These tests provide bacteriological confirmation of TB, which is pathogen-specific and enables definitive diagnosis, case registration, and drug resistance testing” (page 4, lines 54–57 of the marked-up file; page 3, lines 38–39 of the clean file).

Methods –

- *Include how the LOD was calculated.*

This description has been added to the end of the “Single Molecule Array assays” section (page 15, lines 294–296 of the marked-up file; page 10, lines 207–209 of the clean file):

“The limit of detection (LOD) for each assay was calculated as the concentration equivalent to 3 standard deviations above the background, averaged across batches.”

- *In methods section include the range of LAM and Ag85B concentrations used for generating the calibration curves.*

This has been added to the “Single Molecule Array assays” section (page 14, lines 278–280 of the marked-up file; page 10, lines 191–193 of the clean file):

“The concentrations used for generating the calibration curves were between 0.00256 pg/mL and 8 pg/mL for Ag85B and between 0.256 pg/mL and 800 pg/mL for LAM.”

- *The paragraph describing selection of cohorts includes different numbers for training/validation/test in different parts of the paragraph, eg, blinded test cohort is variably described as 217 or 244 participants. This is explained in Figure S9, where again this number is different (215 samples) but could be clearer in the text. In the results section the number for this cohort is 215 samples.*

These numbers in the Study Design have been revised to refer to the numbers of samples actually measured in each cohort: “The study consisted of three cohorts: a model-building (training) cohort of 120 participants, a validation cohort of 241 participants, and a blinded test cohort of 215 participants (Table 1)” (page 12, lines 224–225 of the marked-up file; page 8, lines 145–146 of the clean file). The language in Figure S9 has also been clarified. 244 samples were shipped for the test cohort; 27 were found to be repeated aliquots from the training and validation sets, so these were reassigned as described in the Methods, leaving 217 samples. Two samples could not be measured, with Simoa, leading to a final total of 215 for the blinded test cohort.

• *Where repeat testing of the different aliquots of the same specimen was done (as described in methods), were results always concordant?*

These were nearly always concordant as long as they were detectable, as shown in these scatterplots of the concentration in the second versus first aliquot. (In cases where three aliquots were shipped, we have plotted all three possible comparisons—2 vs. 1, 3 vs. 1, 3 vs. 2—in a lighter grey.)

- *Variation in replicate testing for each sample should be described in more detail – for what proportion of samples did additional replicates need to be run because CVs were >20%?*

13%. This has been added to the Statistics and Reproducibility subsection, in a paragraph moved from Study Design and labelled “Replication” (page 16, lines 313–319 of the marked-up file; page 11, lines 226–232 of the clean file). When Simoa image analysis returned an error (as in 5% of samples) or the three replicates had coefficients of variation above 20% (13% of samples), additional replicates were run. HD-X failures impacted 4% of replicates but 5% of samples; we updated the percentage to the latter to parallel the 13% of samples affected by high CVs.

In how many cases were replicates distributed across the positive/negative threshold?

We have added this information to the end of the Methods section: “For 181/215 samples in the test set (84%), all replicates yielded the same predicted diagnosis. Of the 34 (16%) with discordant replicates, the model using the median was correct in 22 (65%), as opposed to 139/181 (77%) of the ones with concordant replicates” (page 17, lines 346–348 of the marked-up file; page 13, lines 259–261 of the clean file).

• I found Figure S9 hard to reconcile with the explanatory text under subheading 2. It appears that the final analysis included in Figure 2 (black model) was trained completely separately from the initial training model (green), and was done unblinded. Presumably this repeated training was needed because the green model had been trained on largely smear positive samples while the test set was smear negative samples. This seems suboptimal – it might have been better to train the model on a mix of smear/pos and neg and then test a similar ‘test’ set in an blinded manner.

The confusion may be due to the cross-validation and splitting to prevent data leakage. The green model was trained once using the training and validation cohorts and evaluated once. The black models are a series of many models, each trained on 4/5 of the complete cohort and evaluated on the remaining 1/5. This means that the training data for each black model overlapped with the green model, but the training procedure was re-run each time. This repeated training was not “because the green model had been trained on largely smear positive samples while the test set was smear negative samples”—it was done to estimate performance while making the best use of the sample cohort. The nested cross-validation for the black models effectively does what you suggest, by slicing the entire cohort and training and testing on random (disjoint) subsets of it.

We added the following description to Figure S9 legend: “Arrows indicate the flow of data through the study, e.g., the green model is trained on the training/validation cohorts and used to predict the blinded test cohort, which is then unblinded for evaluation, whereas the black models use all samples, with each sample allocated to training or evaluation within each outer fold of nested cross-validation.”

Results-

• The rationale for choosing Ag85B and LAM from the 11 Mtb antigens tested in simoa assay (Table S3) in urine is unclear.

Out of 11 *M. tb* antigens tested in urine using Simoa from individuals with and without TB (Tables S3 and S4, Figure S12), Ag85B and LAM were the only

biomarkers detected in over half of the TB-positive samples and exhibited a statistically significant difference between TB and non-TB samples in the discovery cohort.

This explanation was added to the first paragraph of the Results section (page 18, lines 355–357 of the marked-up file; page 13, lines 267–269 of the clean file) and to the Study design section (page 11, lines 205–206 of the marked-up file; page 7, lines 127–128 of the clean file).

- *Could cross reactivity with LAM-like proteins explain the variation between S4, FIND and G3 in detected LAM concentrations?*

We thank the reviewer for this comment and agree that potential cross-reactivity of the different capture antibodies with other glycolipids or lipopolysaccharides present in urine may explain the variation in detected LAM concentrations. For example, Sigal et al. (*JCM* 56, e01338-18, 2018) reported cross-reactivity of different antibody pairs with other mycobacterium species and non-mycobacteria. Specifically, the FIND28/A194-01 pair had a poor specificity of 63% in urine, with some TB-negative samples giving signals as high as 10-fold above the blank signal. The same pair was shown to cross-react with the non-mycobacterial actinomycetes *Nocardia*, *Gordonia*, *Rhodococcus*, and *Tsukamurella*, as well as cross-react with the mycobacteria *M. fortuitum* and *M. smegmatis*. In contrast, the S4-20/A194-01 pair had no cross-reactivity with fast growing mycobacteria and nonmycobacterial actinomycetes, evident by its higher specificity of 97%.

We added this explanation to the Discussion section (pages 27–28, lines 518–528 of the marked-up file; pages 21–22, lines 428–437 of the clean file).

- *Fig S8, SI is referenced in the wrong statement. Move this to one above.*

We added a reference to Figure S8 in the sentence above (page 19, lines 385–386 of the marked-up file; page 14, line 299 of the clean file).

- *Figure 1 add x-axis labels*

In Figure 1, the x-axis labels are “Ag85B [pg/mL]” and “LAM [pg/mL]”.

- *Subhead 2: Give full form of ROC and CV upon first use.*

These have been fixed in the second paragraph of this subsection: ROC (page 22, line 431 of the marked-up file; page 17, line 343 of the clean file) and CV (page 21, line 423 of the marked-up file; page 16, line 336 of the clean file).

- *Figure 2 legend: “shaded purple rectangles in the upper left-hand corner” not right-hand corner.*

Fixed in the Figure 2 caption (page 24, line 462 of the marked-up file; page 19, line 373 of the clean file).

- *Figure 3: Unclear how there are values of LAM and Ag85B in urine that are below the LOD for each antibody? Are these log values – if so, please relabel axes. Add median and interquartile ranges to the dot plots.*

Negative values of LAM and Ag85B were obtained from the linear regression curve, which is the same transformation used to create the model. However, to reduce confusion and simultaneously address other issues, we combined Figures 3 and 4 to a new Figure 3 with box plots. The y-axes are now log-scaled.

- *Figure S13: What do the values mean? Label for capture and detection antibody. If numbers refer to compatibility of capture/detection antibodies, not sure why if FIND28 has not worked with A194-01 detection antibody, why this combination was chosen?*

A more complete caption was added to Figure S13, indicating that these values are signal-to-background ratio for pooled dilute batches of TB and non-TB urine.

The reference in the text was updated to state that these results were combined with prior findings from the literature:

“These antibody pairs for LAM were chosen after extensive cross-testing of available antibodies using pooled urine from individuals with and without TB (Figure S13), combined with prior findings from the literature” (page 18, line 364 of the marked-up file; page 13, line 277 of the clean file).

- *What is meant by C in HIV+C+ and HIV-C+ samples (results)? Presumably culture, but this is not spelt out anywhere.*

This has been clarified in the Study Design subsection of the Methods (page 12, lines 227ff of the marked-up file; page 8, lines 149–150ff of the clean file).

- *The increment in sensitivity of Simoa over Alere seem to be primarily in HIV-patients, whilst sensitivity was similar in HIV-positive patients. Do the authors have a possible explanation for this?*

LAM levels are lower on average in people without HIV, so there is more room for a more analytically sensitive assay to provide improvement. AlereLAM was, to our knowledge, designed and optimized using samples from people with TB, so it is understandable that a new assay could outperform it more easily in a population for which it had not been developed (HIV patients).

Discussion-

• *“FIND28 measures the highest concentrations in urine on average” – is this actual detection of LAM or simply non-specific background, given that the distribution of concentrations is almost the same in TB and non-TB patients (Figure 3)? What is the specificity of each of these monoclonal antibodies for detection of antigen in urine? Include in discussion if known from previous studies.*

Per the reviewer’s comment, we have changed the word “measures” to “reports” (page 27, line 515 of the marked-up file; page 21, line 424 of the clean file).

The reviewer is correct that cross-reactivity produces non-specific background. We added a paragraph on this in the discussion (pages 27–28, lines 518–534 of the marked-up file; pages 21–22, lines 428–443 of the clean file).

• *This statement at the end of the results section: “For HIV-positive patients, the sensitivities of AlereLAM and the multiplex Simoa assay were 23.8% (5/21) and 28.6% (6/21)” – I thought sensitivity in HIV+ was 68% for Simoa? Perhaps this reflects only smear negative samples (hard to tell from paragraph construction)?*

This section was revised to make the comparison groups more explicit (page 23, lines 451–453 of the marked-up file; page 17–18, lines 362–364 of the clean file). AlereLAM compared head-to-head with Simoa in the test cohort, which saw lower performances of both assays because it contained no smear-positive individuals. Simoa sensitivity in HIV+ was 68% overall and 29% in the test cohort, and it was the latter number for which we could make a direct AlereLAM comparison.

• *For costing, I suspect that the \$6 quoted per replicate reflects actual reagent costs and not the true cost that the assay would be offered at commercially, given its complexity and need for a margin. A more realistic costing might be useful.*

We agree. This back-of-the-envelope estimate would be sensitive not only to number of replicates, but also to economies of scale, capital equipment amortization, profit margin, and the frequent changes in prices of biotech products. However, in light of the editor’s request to foreground negative results, we chose to remove the reference to cost.

- *The test format seems relatively complex and there is a need for complex instrumentation. It seems unlikely that, in this format, the test could be implemented at or close to POC in LMIC.*

We agree with the reviewer that in the current format, the test cannot be implemented as a POC in LMIC. We now discuss this limitation and how the assay can be simplified in the Discussion section (page 33, lines 646–652 of the marked-up file; page 26, lines 521–527 of the clean file).

- *I was confused by this statement: “In addition, the ability to detect 96% of the “likely subclinical” samples as negatives may suggest that our assay can be used in the future as a confirmatory test for this population.” Presumably in this population you want to identify these patients as positives, rather than negatives, as they would benefit from treatment. Are you assuming that the majority of those identified as ‘likely subclinical’ did not in fact have subclinical TB (i.e., were true negatives)?*

We agree with the reviewer that this statement was inappropriate. The "likely subclinical" samples (culture-negative at baseline, culture-positive at follow-up) present interpretive challenges: without knowing when infection occurred, they could represent either true negatives at baseline or early disease that standard culture failed to detect. We pre-specified their classification as TB-negative prior to unblinding, which we believe remains a reasonable methodological choice. However, our post-hoc claim that the 96% negative classification validates the assay for this population was unjustified - had we instead detected most as positive, we could equally have claimed this was early detection of TB cases. We have removed the statement.

- *The authors do not provide a rationale for the use of the multiplex assay as there appears to be no added advantage of using the multiplex compared to S4-20 alone based on AUC scores.*

The multiplex assay was chosen based on noninferiority: although its performance was essentially the same as S4-20 alone (within experimental uncertainty), it may provide more discrimination in the larger, more diverse full cohort. A secondary motivation was scientific, as the development of this multiplex assay allowed us to compare LAM concentrations as measured by different antibody pairs but with the same antigen standard in a large cohort.

Reviewer #2 (Remarks to the Author):

I read with interest study by anonymous authors entitled “Development and validation of a multiplexed single-molecule array urine test for tuberculosis: A case-control diagnostic accuracy study”.

Subject of the study is clearly of interest and worth exploring given a recent rise in TB cases in many settings and well known challenges associated with TB disease diagnosis especially in resource constrained settings, where lack of access to healthcare and timely diagnosis is exacerbated by intrinsic TB laboratory diagnosis problems in those living with HIV.

While the study is generally technologically sound, I think it has quite a few weaknesses on medical and programmatic sides highlighted below. I also felt that study will benefit from re-structuring and significant shortening as follows:

- There are no page and/or line numbers which makes reviewing the paper and making any comments extremely challenging.

We apologize for the inconvenience. We have now numbered the pages and the lines for ease of reviewing.

- Introduction sections spans over 6 pages which is way too long. Will benefit from substantial editing to remove unnecessary details (like performance of different diagnostic assays etc) leaving up to 2 pages max;

We agree with the reviewer and have shortened the Introduction section to 3.5 pages.

- I am not clear on how specificity was calculated in different cohorts. Some cohorts included “clinical TB” specimens – have these been included in specificity and/or sensitivity calculations? What about latent TB specimens? Figure 2 does not contain these subsets and while some results are provided, I would like to see these in one of tables with clearer explanations.

We have added a table to the SI (Table S2) detailing the different categories and classifications. In addition, we have added explanations of the different categories to the Methods section (page 12, lines 227–234 of the marked-up file; page 8, lines 148–155 of the clean file). Most of the requested information is also given in Table 1, which breaks down the numbers of samples with TB (including smear and culture status) and without TB (including latent TB).

- I am not sure what section “Ag85B and LAM concentrations in urine” adds and how this could be interpreted. Ultimately Figures 3 and 4 show that concentrations of all analytes tested (Ag85B and three LAMs) overlap significantly in TB vs non-TB samples, and also vary significantly across settings. Why concentrations of analytes

are so different (up to statistical significance) across different settings? How would this affect assay performance in different settings? Does it mean that assay may work, for instance, in Vietnam, but less likely so in Peru?

We combined Figures 3 and 4 to a new Figure 3 with box plots (page 26 of the marked-up file; page 20 of the clean file). We added a description to the Results meant to clarify the relationship between these sources of variance and assay performance (page 25, lines 480–489 of the marked-up file; pages 19–20, lines 390–399 of the clean file) and updated the subsection title to include “Effects of country of origin, HIV status, and antigen heterogeneity” (page 24, lines 473–474 of the marked-up file; page 19, lines 384–385 of the clean file). In keeping with the editor’s request to reframe the manuscript to focus on negative results, we addressed possible causes and implications for this variability according to HIV status (page 28, lines 528–534 of the marked-up file; page 8, lines 148–155 of the clean file) and country (page 31, lines 599–612 of the marked-up file; page 24, lines 495–506 of the clean file).

Our assay’s performance is validated in all of these countries, and is thus robust to this variation. However, it could have appeared more accurate if we had only used samples from Vietnam, which showed the greatest separation between TB and non-TB. We cannot be certain whether this is a feature of Vietnam’s health system, of the TB strains circulating in Southeast Asia, or any other generalisable factor, or specific to the clinic from which FIND collected specimens and its catchment.

- I cannot fully understand last paragraph of Results section. Authors tested their assay in parallel with commercial Alere LAM assay on two subsets of samples comprising 215 and 244 specimens. What are these samples? Why sensitivity of multiplex Simoa in samples collected from PLHIV was significantly lower in these experiments compared to claimed sensitivity in overall cohort (28.6% in a head to head comparison vs 58% in training and validation cohorts)?

The 244 samples consist of all samples *sent* in the final shipment for the test cohort; Of those, 215 samples were used as the blinded test cohort. The additional 29 samples were found to be repeated aliquots from the training and validation sets; therefore, they were not used for the blinded cohort (results from these duplicate aliquots were combined and assigned to the earliest cohort in which the sample appeared, in order to preserve blinding), but were used for the AlereLAM comparison. Sensitivity is lower because this cohort was all smear-negative.

This paragraph has been revised to make the relationship of these overlapping cohorts clearer. We rephrased this as 29 additional samples from the training and validation cohorts (page 22, lines 447–448 of the marked-up file; page 17, lines 358–359 of the clean file) and clarified that sensitivity is lower because this cohort was all smear-negative (page 23, lines 451–453 of the marked-up file; pages 17–18, lines 362–364 of the clean file).

- What are the interpretation rules of the multiplex assay? I mean, how are Ag58B quantification results combined with LAM detection results?

The multiplex assay concentrations are transformed into a TB result according a generalized additive model (GAM), as described in the Methods (page 16, lines 326–328 of the marked-up file; page 12, lines 239–241 of the clean file) and Results (page 21, lines 414–432 of the marked-up file; pages 16–17, lines 327–344 of the clean file). Our description of this approach in the Results section has been expanded (page 21, lines 416–420 of the marked-up file; page 16, lines 329–333 of the clean file).

- Authors claim that their assay is more sensitive in PLHIV compared to the only commercially available assay Alere LAM. Firstly, I would like to see more recent references, for example Adzemovic et al., CID 2025 where results of a latest version of Fujifilm TB LAM assay evaluation in Uganda were published; assay has not yet been endorsed by WHO but it's claimed to employ an ultrasensitive LAM detection technology so it's worth including in in discussion. Secondly, I would like to see considerations regarding potential commercialization of Simoa assays in terms of technology and prices – at the moment Simoa seems too difficult to commercialise which should be considered in Discussion section to put in the context of unmet medical needs, cost effectiveness and health technology assessment.

Mention of Adzemovic et al., CID 2025 has been added to the Discussion (page 29, lines 563–567 of the marked-up file; page 23, lines 460–463 of the clean file). Although we have retained considerations of translation to the clinic (page 33, lines 646–652 of the marked-up file; page 26, lines 521–527 of the clean file), we elected not to include a revised cost estimate (also suggested by Reviewer #1) due to the shift of overall focus to negative results and this work's lesson's for future LAM and related assay development.

Reviewer #3 (Remarks to the Author):

I co-reviewed this manuscript with one of the reviewers who provided the listed reports. This is part of the Communications Medicine initiative to facilitate training in peer review and to provide appropriate recognition for Early Career Researchers who co-review manuscripts.

Reviewer #2 (Remarks to the Author):

I would like to thank authors for addressing my and other reviewers' comments. While the manuscript has benefited from the modifications, I still think there is a room for improvements and further clarifications as follows:

The work's primary interest is in showing that adding additional antibody pairs for LAM does not substantially improve test accuracy, supporting the notion that LAM in the urine of many TB patients may be below the level of detection of a highly sensitive assay, and that further work along these lines may not demonstrate substantial improvements in sensitivity.

- Introduction section is still too heavy. Suggest further redacting it and remove or significantly reduce fragments related to LAM (lines 112 -133).

The section in the Introduction related to LAM was rewritten to be more concise. However, per the editor's request, we added a short paragraph at the end of the Introduction to further summarize key findings.

- I am concerned about classifying subclinical TB as negative (i.e. non-TB, line 234) for the purposes of the assay validation. Author mention that those with subclinical TB had changes on their CXR suggestive of TB and also had TB cultures on subsequent visits. This essentially means that those with subclinical TB had an active TB (vs latent TB infection) and SHOULD be detected using rapid assays including any LAM- or Ag85B based tests.

Author report that 26/27 (96%) samples from patients with subclinical TB tested negative on their assay translating into sensitivity for subclinical TB of just 3.7%. This require explanations that need to be added to Discussion section.

We have revised the Methods to clarify the basis for classifying "likely subclinical TB" cases as non-TB (page 9, lines 183–189 in the marked-up file; pages 7–8, lines 143–148 in the clean file). This category is defined by FIND based on culture positivity at follow-up (44–175 days after enrollment), not baseline characteristics—in contrast to usual definitions of "subclinical TB" in the literature. All cases were bacteriologically negative at enrollment by multiple tests. We cannot distinguish baseline culture-negative TB from incident infection acquired after enrollment, so we classified these cases according to their baseline status. We have also corrected Table S5 to reflect that some had normal chest X-rays and all had symptoms.

- I am also not convinced with explanations on why performance of the assay differs across settings. Prevalence of different genetic families/lineages so differ across settings but chemical composition of LAM/Ag85B antigens do not. Please try to provide some evidence on why lineages may be associated with varying concentrations of antigens in urine - to be honest, I cannot come up with such an explanation.

We thank the reviewer for their criticism. We hedged the text of this section to emphasize that we cannot explicitly determine the origin of the heterogeneity seen across countries. However, we suggest several possibilities. We also found references that provide evidence that LAM composition differs across lineages.

We added and revised the Discussion as follows (pages 26–27 in the marked-up file; page 25 in the clean file):

Significant heterogeneity was observed in biomarker concentrations across countries (Figure 3). Although we cannot explicitly determine the origin of this heterogeneity, we suggest several possibilities. One possible explanation is the different time of sampling across countries. - In some countries, there are extreme delays in patients seeking treatment, which is also one of the major hurdles for controlling TB. For example, a four-week delay in seeking treatment was seen in KwaZulu Natal, South Africa, and a ten-week delay in South Africa's rural Northern Province, now officially known as Limpopo, which far exceeds the WHO-recommended two weeks for initiating treatment after suspicion ³⁸. Reasons for delay include distance from diagnostic facilities, long waiting times, and absence of clinical symptoms ^{38,39}. Delays in diagnostic testing in different countries can be part of the cause for lower concentrations of biomarkers in the urine at the time of sampling. It may be valuable to record the number of days from the onset of symptoms to seeking a diagnosis, especially when acquiring samples for clinical trials, as the concentration of biomarkers can be affected. (FIND collected a “binned” version of symptom duration for some individuals in our study, but the data were too coarse for this type of analysis.) In addition, this information may aid in assessing pre-test probabilities.

Another explanation for the heterogeneity in biomarker concentrations across countries may be attributed to different LAM compositions in different lineages, some of which is explained by country of origin (Figure 3). The higher concentrations of biomarkers in TB+HIV+ individuals from Vietnam may be attributed to the different lineages in each country. *M.tb* has seven lineages: i. In South Africa and Peru, the

dominant lineage is lineage 4, in Vietnam lineages 1 and 2 ^{40,41}, and in Cambodia lineage 1 ⁴². There is evidence that the relative Mannose capping of LAM differs across lineages ⁴³. The differences in LAM capping can result in different binding of the antibodies to LAM, and therefore, affect the measured concentrations in urine. the dominant epitopes present in each country may imply that different LAM forms are present in the urine of patients from different countries with different lineages. The overlap between the concentrations of Ag85B and LAM in non-TB and TB+ individuals in Peru could indicate ~~a different health system collection procedure or different culture protocols, affecting the classification of these samples.~~ Therefore, ~~to ensure the assay works in all settings, standardization of sample collection and sample classification would be necessary.~~ Another option is that in Peru, other mycobacteria are present to which the antibodies bind. In future studies, such as the prospective validation of the present assay, it would be instructive to conduct mycobacterial sequencing and compare biomarker abundance results with lineage and phenotype information.